# Nomic Embed: Training a Reproducible Long Context Text Embedder

**Zach Nussbaum**                                                      *zach@nomic.ai*
*Nomic AI*

**John X. Morris**                              *jack@nomic.ai, jxm3@cornell.edu*
*Nomic AI, Cornell University*

**Brandon Duderstadt**                                        *brandon@nomic.ai*
*Nomic AI*

**Andriy Mulyar**                                              *andriy@nomic.ai*
*Nomic AI*

**Reviewed on OpenReview:** *https://openreview.net/forum?id=IPmzyQSiQE*

## Abstract

This technical report describes the training of nomic-embed-text-v1, the first fully reproducible, open-source, open-weights, open-data, 8192 context length English text embedding model that outperforms both OpenAI Ada-002 and OpenAI text-embedding-3-small on the short-context MTEB benchmark and the long context LoCo benchmark. We release the training code and model weights under an Apache 2.0 license. In contrast with other open-source models, we release the full curated training data and code that allows for full replication of nomic-embed-text-v1. You can find code and data to replicate the model at https://github.com/nomic-ai/contrastors.

## 1 Introduction

Text embeddings are an integral component of modern NLP applications powering retrieval-augmented-generation (RAG) for LLMs and semantic search (Lewis et al., 2021a; Izacard et al., 2022b; Ram et al., 2023). These embeddings encode semantic information about sentences as low-dimensional vectors that are used in downstream applications, such as clustering for data visualization, classification, and information retrieval.

The majority of the top open-source models on the MTEB benchmark (Muennighoff et al., 2023) are limited to context lengths of 512, such as E5 (Wang et al., 2022), GTE (Li et al., 2023), and BGE (Xiao et al., 2023). This short context length reduces model utility in domains where the overall document semantics are not localized to sentences or paragraphs. Most top embedding models with a context length longer than 2048 are closed-source, such as Voyage-lite-01-instruct (Voyage, 2023) and text-embedding-ada-002 (Neelakantan et al., 2022).

As of October 2024, the top-performing open-source long context embedding models are jina-embedding-v2-base-en (Günther et al., 2024) and E5-Mistral-7b-instruct (Wang et al., 2023b). Unfortunately, jina-embedding-v2-base does not surpass OpenAI's text-embedding-ada-002 (Neelakantan et al., 2022) (see Table 1). Further, E5-Mistral (Wang et al., 2023b) is not feasible to use in many engineering applications due to the large inference requirements of a 7 billion parameter transformer, and does not perform well beyond 4096 tokens.

In this paper, we present an end-to-end training pipeline for a state of the art long context text embedding model at only 137 million parameters. nomic-embed-text-v1 outperforms OpenAI text-embedding-ada and

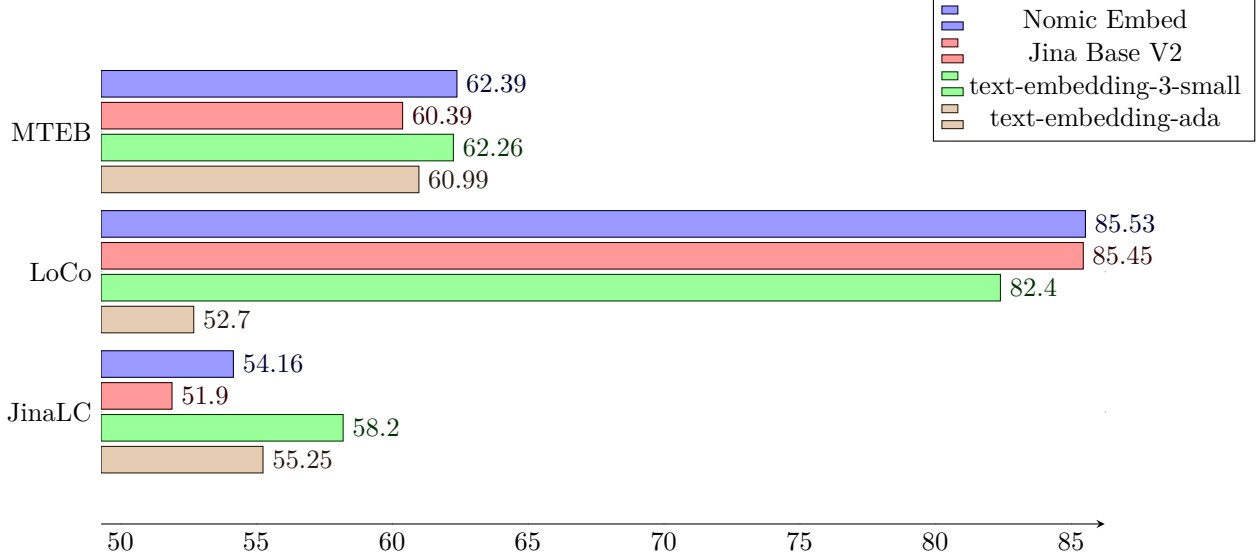

Figure 1: **Benchmarking Text Embedding Model.** Aggregate performance of nomic-embed-text-v1, OpenAI text-embedding-ada, OpenAI text-embedding-3-small and jina-embedding-base-v2 on both short and long context benchmarks. nomic embed is the only fully auditable long context model that exceeds OpenAI text-embedding-ada and OpenAI text-embedding-3-small on MTEB and LoCo. nomic embed performs similarly or outperforms Jina Base V2 on all tasks. X-axis units vary per benchmark suite.

text-embedding-3-small performance on short context (MTEB) and long context benchmarks (LoCo) (Table 1).

Further, we are the first to release all training artifacts needed to train a high-performing text embedding model. We release the model weights, training code, and training data to enable end-to-end auditability and replication of the model

## 2    Related Work

Text embedding models have historically been trained with sequence lengths less than or equal to 512 tokens. Recently, Günther et al. (2024) trained a long context text embedding model, jina-embeddings-base-v2, but underperforms closed source text embedding models like text-embedding-ada-002 on both the MTEB benchmark as well as the Jina Long Context Benchmark. Additionally, jina-embeddings-base-v2 underperforms other open-weight short-context text embedding models like E5 (Wang et al., 2022), GTE (Li et al., 2023), and BGE (Xiao et al., 2023).

Further, there is a lack of transparency into the training pipeline for high performing open-weight text embedding models. Many of these released models omit key details such as data source, data curation techniques, and training code. Wang et al. (2022) outlined their data filtering procedure for E5 which includes first training a model over a large noisy dataset and then using the resulting model to filter low quality text pairs. However, they do not release details on the model used for consistency filtering, how it was trained, or what data was used. They additionally do not release any training code or data for the released embedding model. Similarly, Li et al. (2023) and Günther et al. (2024) did not detail the data sources for contrastive pretraining, omit details on how data filtering and mining was approached, and did not release training code.

Additionally, few details have been released on how closed source text embedding models are trained like Voyage-lite-01-instruct (Voyage, 2023) and OpenAI's text-embedding-ada-002 and text-embedding-3 (Neelakantan et al., 2022).

# 3  Background

State-of-the-art text embedding models are generally trained in three stages: masked language modeling (Devlin et al., 2019), weakly-supervised contrastive pretraining, and contrastive finetuning (Wang et al., 2022; Li et al., 2023; Günther et al., 2023; 2024). Traditionally, finetuning involved leveraging labeled datasets such as MSMarco and SNLI (Bowman et al., 2015) to generate paired training data for the contrastive signal. Examples include SBERT (Reimers & Gurevych, 2019), SimCSE (Gao et al., 2022), and SGPT (Muennighoff, 2022). Recent models such as E5 (Wang et al., 2022), GTE (Li et al., 2023), BGE (Xiao et al., 2023), InstructOR (Su et al., 2023a), and Jina (Günther et al., 2023; 2024) utilize a multi-stage regime in which a pretrained transformer is first contrastively trained using a large corpus of weakly paired data (e.g. Quora, Reddit Comments) and then additionally finetuned on small, higher quality labeled datasets such as MSMarco. The two-stage paradigm significantly improves model quality as weakly paired data is available in much greater quantity.

Table 1: nomic-embed-text-v1 is the only open-source long-context model to outperform closed source models like text-embedding-ada-002 and text-embedd-3-small on the short-context MTEB benchmark and the long context LoCo benchmark.

| Model | Params | Seq | MTEB | LoCo | Jina LC | Weights | Code | Data |
|---|---|---|---|---|---|---|---|---|
| nomic-embed-text-v1 | 137M | 8192 | **62.39** | 85.53 | 54.16 | **Yes** | **Yes** | **Yes** |
| nomic-embed-text-v1-ablated | 137M | 8192 | 61.36 | **86.89** | 53.53 | **Yes** | **Yes** | **Yes** |
| jina-embeddings-base-v2-en | 137M | 8192 | 60.39 | 85.45 | 51.90 | **Yes** | No | No |
| text-embedding-ada-002 | N/A | 8192 | 60.99 | 52.70 | 55.25 | No | No | No |
| text-embedding-3-small | N/A | 8192 | 62.26 | 82.4 | **58.21** | No | No | No |
| E5-Mistral-7b-instruct | 7B | 4096 | 66.6 | 87.8 | N/A | Yes | No | No |
| text-embedding-3-large | N/A | 8192 | 64.59 | 79.4 | 58.69 | No | No | No |

## 3.1  Masked Language Modeling

Masked language modeling masks a percentage of inputs and trains a bidirectional transformer to predict the masked tokens (Devlin et al., 2019). The original BERT model was additional trained with a binary auxiliary Next-Sentence Prediction (NSP) task. Liu et al. (2019) released RoBERTa in which they attained better performance by training on more data and for longer. They additionally removed the NSP task as it didn't show any performance improvements. More recently, Portes et al. (2023) introduced MosaicBERT, an efficient and high performing BERT training recipe by increasing the masking rate, incorporating FlashAttention (Dao et al., 2022), and other training optimizations.

## 3.2  Weakly-supervised Contrastive Pretraining

Weakly-supervised contrastive pretraining aims to teach a model to distinguish the most similar documents from other irrelevant documents. To do so, we employ the InfoNCE contrastive loss (van den Oord et al., 2019). For a given batch $B = (q_0, d_0), (q_1, d_1), ..., (q_n, d_n)$, we minimize the loss function:

$$\mathcal{L}_C = -\frac{1}{n} \sum_i \log \frac{e^{s(q_i, d_i)/\tau}}{e^{s(q_i, d_i)/\tau} + \sum_{j \neq i}^{n} e^{s(q_i, d_j)/\tau}} \tag{1}$$

where $s(q, d)$ is the (learned) score between query $q$ and document $d$. We set $s$ to cosine similarity for all of our experiments. Contrary to other approaches, we adopt a unidirectional contrastive loss from query to document. Other approaches like Günther et al. (2023) use a bidirectional contrastive loss by including the contrastive loss from document to query as well.

### 3.3 Contrastive Finetuning

The last stage of training aims to boost performance by utilizing human-labeled datasets. Several papers including Ni et al. (2021a;b); Wang et al. (2022); Li et al. (2023) have shown that finetuning on these datasets leads to improvements in downstream performance, especially for QA and web-search retrieval tasks. We adapt Equation 1 to include hard negative documents in each batch:

$$\mathcal{L}_C = -\frac{1}{n} \sum_i \log \frac{e^{s(q_i,d_i)/\tau}}{e^{s(q_i,d_i)/\tau} + \sum_{j \neq i}^{n} e^{s(q_i,d_j)/\tau} + \sum_{m=1}^{H} e^{s(q_i,d_{hn}(1,m))/\tau}} \tag{2}$$

Here, we modify the partition function of the contrastive loss to include $H$ hard negative documents $d_{hn}(1,m)$ which are documents specially chosen to be close to $d$ but not true positive documents of $q$.

### 3.4 Rotary Positional Embeddings

Rotary Positional Embeddings (RoPE) are an alternative positional encoding introduced in (Su et al., 2023b) that encode relative positional information through rotations within the attention-layers.

Following the notation in (Su et al., 2023b), setting $\theta = 10,000$ is standard. We set $\theta = 1,000$ for our experiments but found little to no performance degradation.

### 3.5 RoPE Context Length Extrapolation

However, a limitation with RoPE is in scaling to sequence lengths longer than seen during training. We discuss two methods: position interpolation and frequency-based scaling.

#### 3.5.1 Position Interpolation

Chen et al. (2023) and kaiokendev (2023) independently proposed extrapolating RoPE based models by interpolating the position indices to be within the original training sequence length. Following the notation in (Su et al., 2023b), given a pretrained model with context length L, the position embedding function $f_W$ is modified as:

$$f_W(x_m, m, \theta_d) = f_W(x_m, \frac{mL}{L'}, \theta_d) \tag{3}$$

where $L' > L$ is the target extended context length. This approach, while simple, requires fine-tuning on a smaller dataset to achieve stable performance at longer contexts.

#### 3.5.2 Frequency-based Scaling

**NTK-Aware Scaling**  bloc97 (2023) first proposed scaling the high frequencies more and low frequencies less by changing the base $\theta$ in order to be "NTK-aware" as it was shown in Tancik et al. (2020) that neural networks struggle to represent high frequencies well. The base is scaled by the ratio of the longer sequence length and the trained sequence length:

$$b^` = b * s^{\frac{|D|}{|D|-2}} \tag{4}$$

where $s = \frac{L^`}{L}$.

**Dynamic NTK Scaling**  Dynamic NTK scaling (emozilla, 2023; Peng et al., 2023) improves upon NTK-aware by introducing a hyperparameter $\alpha$ to Equation 4.

$$b^` = b * ((\alpha * s) - (\alpha - 1))^{\frac{|D|}{|D|-2}} \tag{5}$$

This maintains the original position embeddings for sequences within the pretrained context length ($l_{current} \leq L$) and gradually scales the embeddings as sequences grow longer, preventing abrupt performance degradation. Additionally, this approach can be used without any finetuning as shown in (Peng et al., 2023).

## 4    Methods

### 4.1    Masked Language Modeling

#### 4.1.1    Data

Following Devlin et al. (2019), we use BooksCorpus (Zhu et al., 2015) and a Wikipedia dump from 2023 to train a long context BERT model, hereinafter called nomic-bert-2048. Each document from BooksCorpus and Wikipedia is tokenized using the bert-base-uncased tokenizer from Devlin et al. (2019) and packed across documents to chunks of 2048 tokens. If a document is shorter than 2048 tokens, we append another document until it fits 2048 tokens. If a document is greater than 2048 tokens, we split it across multiple documents. nomic-bert-2048 follows a similar training pipeline for masked language modeling as Portes et al. (2023). We omit next sentence prediction similarly to Liu et al. (2019) and Portes et al. (2023) as it was shown to not improve performance and simplifies the training recipe.

#### 4.1.2    Training Modifications

To train a long sequence length and efficient BERT, we adapt the BERT architecture. We make the following architecture changes to BERT base (Devlin et al., 2019):

- Substituting absolute positional embeddings for rotary positional embeddings (Su et al., 2023b)

- Using SwiGLU activation instead of GeLU (Shazeer, 2020)

- Using Flash Attention (Dao et al., 2022)

- Setting Dropout to 0 (Geiping & Goldstein, 2022)

- Vocab size as a multiple of 64 (Portes et al., 2023; Shoeybi et al., 2020)

resulting in a 137 million parameter encoder.

We train all stages with a max sequence length of 2048 and employ Dynamic NTK interpolation at inference to scale to 8192 sequence length (Peng et al., 2023; emozilla, 2023). Additionally, we opt for SwiGLU versus GeGLU like proposed in Portes et al. (2023) as runtime is roughly 25% faster for SwiGLU using the Flash Attention repository[1].

We use a 30% masking rate instead of 15% following Portes et al. (2023) and we remove the Next Sentence Prediction task to simplify the training recipe (Liu et al., 2019; Portes et al., 2023). We use the AdamW optimizer (Loshchilov & Hutter, 2019) with a max learning rate of 5e-4 with $\beta_1 = 0.9$ $\beta_2 = 0.98$. We employ a linear warmup of 6% of the total training steps and a linear decay to 0. We use a global batch size of 4096 with gradient accumulation over 8 batches. We utilize DeepSpeed (Rajbhandari et al., 2020) stage 2 to fit larger batches into memory. Additionally, we use bfloat16 mixed precision and fp32 for gradient accumulation dtype. We disable gradient clipping (Liu et al., 2019) and set weight decay to 1e-5. We call our final model nomic-bert-2048 and also release its weights.

### 4.2    Weakly-Supervised Contrastive Pretraining

#### 4.2.1    Data

Similar to Wang et al. (2022); Li et al. (2023); Xiao et al. (2023); Ni et al. (2022), we use large collections of publicly available data to form contrastive pairs. These datasets span various objectives and domains, from web retrieval to clustering of scientific articles. In total, we curated 470 million pairs across 29 datasets[2].

**Consistency Filtering**: Since many of these datasets may contain noisy examples, we employ consistency filtering to remove the potential false positives in the dataset (Günther et al., 2023; Wang et al., 2022).

---

[1] https://github.com/Dao-AILab/flash-attention/tree/main
[2] https://huggingface.co/datasets/sentence-transformers/embedding-training-data

Consistency filtering uses a pretrained model to filter out potential noisy examples in an effort to improve data quality and subsequently model quality. Additionally, reducing the total number of examples needed to train a high quality text embedding model can reduce the overall cost to train the text embedding model.

For each pair, described as (*query*, *document*), we embed the queries and documents separately. We sample 1 million points from the dataset and for each query, we find the top-k (in this case 2) neighbors using cosine similarity. If *document* is not in the top-k neighbors, we discard the example.

Günther et al. (2023) uses all-MiniLM-L6-v2[3], a 22 million parameter sentence embedding model for consistency filtering. However, we found that it regularly discarded retrieval pairs that were true positives but had low lexical overlap. We instead utilized gte-base[4] (Li et al., 2023), a 109 million parameter model for consistency filtering. After filtering, we end up with ∼235 million pairs. The full dataset distribution can be seen in Appendix B.

We additionally explored consistency filtering using a cosine similarity threshold instead of the method described above. For a given pair, if the cosine similarity was greater or equal to the threshold, we kept the pair and otherwise discarded. However, we abandoned this approach in favor of top-k consistency filtering as we found through manual inspection the threshold consistency filtering discarded high quality retrieval pairs that had low cosine similarity. We additionally noticed lower retrieval scores in models trained using a threshold for consistency filtering versus using top-k consistency filtering.

**Curating Long Context Text Pairs**: As the majority of these datasets are composed of sequences shorter than 2048 tokens we additionally curate long context datasets to allow for the learning of long-range dependencies. We use Wikipedia titles paired with the corresponding body and S2ORC (Lo et al., 2020) abstracts and full paper text from a single paper.

Table 2: nomic-bert-2048 performs similarly to other short and long-context encoders when evaluated on the GLUE benchmark.

| Model | Seq | Bsz | Steps | Cola | SST2 | MRPC | STSB | QQP | MNLI | QNLI | RTE | Avg |
|---|---|---|---|---|---|---|---|---|---|---|---|---|
| MosaicBERT | 128 | 4k | 178k | 0.59 | 0.94 | 0.89 | 0.90 | 0.92 | 0.86 | 0.91 | 0.83 | 0.85 |
| JinaBERTBase | 512 | 4k | 100k | 0.51 | 0.95 | 0.88 | 0.90 | 0.81 | 0.86 | 0.92 | 0.79 | 0.83 |
| RobertaBase | 512 | 8k | 500k | 0.64 | 0.95 | 0.90 | 0.91 | 0.92 | 0.88 | 0.93 | 0.79 | 0.86 |
| MosaicBERT | 2k | 4k | 70k | 0.54 | 0.93 | 0.87 | 0.90 | 0.92 | 0.86 | 0.92 | 0.82 | 0.85 |
| nomic-bert-2048 | 2k | 4k | 100k | 0.50 | 0.93 | 0.88 | 0.90 | 0.92 | 0.86 | 0.92 | 0.82 | 0.84 |

You can access the training data of nomic-embed-text-v1 by visiting the code repository . You can explore a 5M sample of our contrastive training pairs at https://atlas.nomic.ai/map/nomic-text-embed-v1-5m-sample.

### 4.2.2 Training Modifications

We initialize the model for weakly-supervised contrastive training with the weights of nomic-bert-2048. We use a global batch size of 16,384. We use AdamW with a learning rate of 2e-4, $\beta_1 = 0.9$, $\beta_2 = 0.999$, and weight decay of 0.01. Gradient clipping is set to 1.0. We use a linear warmup schedule of 700 steps and an inverse square root decay schedule.

We sample one data source and fill each batch with only data from that source to discourage the model learning source-specific shortcuts. We train with a max sequence length of 2048 for 1 full epoch over the weakly-supervised contrastive data. Full details on data composition can be found in Appendix B.

Due to GPU memory constraints, we employ GradCache (Luyu Gao & Callan, 2021) as well as mixed precision training (Micikevicius et al., 2018).

---

[3]all-MiniLM-L6-v2 model https://huggingface.co/sentence-transformers/all-MiniLM-L6-v2
[4]gte-base model (https://huggingface.co/thenlper/gte-base)

Finally, we use task specific prefixes to break the symmetry of the biencoder as in Wang et al. (2022). Without prefixes, the model receives conflicting reward signal. Consider the case of determining which document is closest to the query "What is the capital of France?":

1. "What is the name of the capital city of France?

2. "Paris is the capital of France."

A semantic similarity task would consider the first closest, while a question answering task would consider the second closest. Prefixes enable the model to distinguish between the behaviors specified by each of these tasks.

We use the following task-specific prefixes:

- `search_query`

- `search_document`

- `classification`

- `clustering`

inspired by Reimers et al. (2023). We first break prefixes into two categories: symmetric, where the query and document have a similar structure, and asymmetric, where the query is usually a single sentence and the document can be many sentences (Su et al., 2023a). The first two prefixes are used for retrieval tasks: where `search_query` is used for the question and `search_document` is used for the response. `classification` is used for STS-related tasks like rephrasals. `clustering` is used for tasks where the objective is to group semantically similar texts close together, like Arxiv title-abstract pairs. For symmetric tasks, the same prefix is appended to both the query and document.

### 4.2.3 Supervised Contrastive finetuning

### 4.2.4 Data

Supervised fine tuning is performed on MSMarco (Bajaj et al., 2018; Wang et al., 2023a), NQ (Karpukhin et al., 2020; Gao & Callan, 2021), NLI (Gao et al., 2022), HotpotQA (Yang et al., 2018), FEVER (Thorne et al., 2018), portions of MEDI (Su et al., 2023a), WikiAnswers (Fader et al., 2014), and Reddit[5].

For the datasets MSMarco, NQ, NLI, FEVER, and HotpotQA, we train over the released training sets from the BEIR benchmark (Thakur et al., 2021). For the retrieval datasets (MSMarco, NQ, HotpotQA, and Fever), we mine negatives, if not already mined, using gte-base (Li et al., 2023). For each $(q, d)$ pair, we find the top 20 documents among the corpus most similar to the query $q$, excluding $d$, and use these as hard negatives. For other non-retrieval datasets, we randomly sample negatives among the corpus in place of mining hard negatives as we found that mining did not improve performance.

Although the BEIR component of MTEB was originally intended as a zero shot benchmark, several open source models, such as those in Xiao et al. (2023); Li et al. (2023); Wang et al. (2023b), report training on train splits of BEIR benchmark datasets such as FEVER and HotpotQA. We report results for nomic-embed-text-v1-ablated trained without FEVER, HotpotQA, and MEDI.

Similarly to the weakly supervised contrastive stage, we sample a dataset and fill a batch with all points from that chosen dataset. In total, we train on 1.6 million datapoints. The full dataset distribution can be seen in Table 3.

---

[5]https://github.com/PolyAI-LDN/conversational-datasets/tree/master/reddit

Table 3: Supervised finetuning dataset distribution.

| Dataset | Number of Samples |
|---|---|
| MSMarco | 484,864 |
| NLI | 275,200 |
| Reddit | 199,680 |
| Medi Supernli | 177,408 |
| Hotpot | 169,728 |
| Fever | 139,776 |
| Medi Stackexchange | 100,352 |
| NQ | 69,888 |
| Medi Flickr | 50,944 |
| Medi Wiki | 24,832 |

### 4.2.5 Training Modifications

We train for one epoch using seven hard negatives per pair and a batch size of 256. We employ a learning rate of 2e-5, $\beta_1 = 0.9$, $\beta_2 = 0.999$, and weight decay of 0.01. Gradient clipping is set to 1.0. We use a linear warmup schedule of 400 steps and a linear cooldown to 0 and train with prefixes as described above. We found that increasing the number of negatives above 7 does not significantly improve performance. We also found that training for multiple epochs hurts performance. Instead of choosing the first N negatives, we randomly sampled the mined negatives. We found this to improve performance as some of the mined negatives introduced false negatives.

## 5 Results

### 5.1 nomic-bert-2048 GLUE Results

We first evaluate nomic-bert-2048 on the GLUE benchmark (Wang et al., 2019) to verify that our adapted BERT architecture performs similarly or better compared to similar encoders. The GLUE benchmark consists of 9 tasks, but we evaluate on 8 similar to Liu et al. (2019). We follow the evaluation methodology presented in Liu et al. (2019). Roberta numbers are taken from Table 8 in (Liu et al., 2019). MosaicBert numbers are taken from Table S1 in Portes et al. (2023) except for the 2048 model which we evaluated in the same manner as nomic-bert-2048. JinaBertBase Glue Test numbers reported in Table 2 from (Günther et al., 2024).

For each task, we train for 10 epochs with batch sizes 16, 32 and learning rate 1e-5, 2e-5, 3e-5 with a linear warmup of 6% across 5 seeds. The median score per task at the end of the 10 epochs is presented in Table 2. Note we report accuracy for MRPC and QQP and Pearson for STSB [6]. Similar to Liu et al. (2019), we initialize from an MNLI checkpoint for RTE, STSB, and MRPC.

Across all tasks, nomic-bert-2048 scores similarly to MosaicBERT (Portes et al., 2023) except on Cola. MosaicBERT is trained with more gradient updates on C4 (Raffel et al., 2019). However, nomic-bert-2048 was trained with a longer sequence length and in effect has seen more tokens during pretraining. The difference in results could be due to a few reasons. First the training corpus could lead to better results on Cola as nomic-bert-2048 trains on Wikipedia and Bookscorpus while MosaicBERT trains on C4 which tends to skew towards shorter sequence lengths. Additionally, nomic-bert-2048 utilizes RoPE while MosaicBERT uses ALiBi for long-context extrapolation.

JinaBERT also trains a similar model to MosaicBERT using ALiBI for long-context extrapolation and C4 as its training corpus but sets the max sequence length to 512. It performs slightly worse on average to nomic-bert-2048 and MosaicBERT. Even though it was trained similarly to MosaicBERT, JinaBERT performs worse on Cola, RTE, and QQP. These findings are similar when compared to nomic-bert-2048 except Cola where JinaBERT outperforms nomic-bert-2048.

---

[6]`https://github.com/facebookresearch/fairseq/issues/1561#issuecomment-571729519`

Table 4: MTEB benchmark results (Muennighoff et al., 2023). nomic-embed-text-v1 outperforms all similarly sized models on short-context tasks except BGE-Base.

| Category → Number of datasets → | Params. | Cls. 12 | Clust. 11 | PairCls. 3 | Rerank 4 | Retr. 15 | STS 10 | Summ. 1 | Avg 56 |
|---|---|---|---|---|---|---|---|---|---|
| *Unsupervised Models* | | | | | | | | | |
| Glove (Pennington et al., 2014) | 0.3B | 57.3 | 27.7 | 70.9 | 43.3 | 21.6 | 61.9 | 28.9 | 42.0 |
| SimCSE (Gao et al., 2022) | 110M | 62.5 | 29.0 | 70.3 | 46.5 | 20.3 | 74.3 | 31.2 | 45.5 |
| nomic-embed-text-v1$_{\text{unsup}}$ | 137M | 71.2 | 42.5 | 83.7 | 55.0 | 48.0 | 80.8 | 30.7 | 59.9 |
| *Supervised Models* | | | | | | | | | |
| SimCSE$_{\text{bert-sup}}$ (Gao et al., 2022) | 110M | 67.3 | 33.4 | 73.7 | 47.5 | 21.8 | 79.1 | 23.3 | 48.7 |
| Contriever (Izacard et al., 2022a) | 110M | 66.7 | 41.1 | 82.5 | 53.1 | 41.9 | 76.5 | 30.4 | 56.0 |
| E5$_{\text{base}}$ (Wang et al., 2022) | 110M | 75.2 | 44.2 | 86.0 | 56.6 | 50.6 | 82.1 | 30.2 | 61.6 |
| GTE$_{\text{base}}$ (Li et al., 2023) | 110M | 73.0 | 46.2 | 84.6 | 58.6 | 51.1 | 82.3 | 31.2 | 62.4 |
| BGE$_{\text{base}}$ (Xiao et al., 2023) | 110M | 75.5 | 45.8 | 86.6 | 58.9 | 53.3 | 82.4 | 31.1 | 63.6 |
| Jina$_{\text{v2}}$ (Günther et al., 2024) | 137M | 73.5 | 41.7 | 85.4 | 57.0 | 47.9 | 80.7 | 31.6 | 60.4 |
| nomic-embed-text-v1-ablated | 137M | 73.6 | 43.7 | 84.6 | 53.3 | 51.4 | 80.2 | 31.3 | 61.4 |
| nomic-embed-text-v1 | 137M | 74.1 | 43.9 | 85.2 | 55.7 | 52.8 | 82.1 | 30.1 | 62.4 |
| E5$_{\text{large-v2}}$ (Wang et al., 2022) | 335M | 75.2 | 44.5 | 86.0 | 56.6 | 50.6 | 82.1 | 30.2 | 62.3 |
| GTE$_{\text{large}}$ (Li et al., 2023) | 335M | 73.3 | 46.8 | 85.0 | 59.1 | 52.2 | 83.4 | 31.7 | 63.1 |
| BGE$_{\text{large}}$ (Xiao et al., 2023) | 335M | 76.0 | 46.1 | 87.1 | 60.0 | 54.3 | 83.1 | 31.6 | 64.2 |
| GTR$_{\text{xxl}}$ (Ni et al., 2021a) | 4.8B | 67.4 | 42.4 | 86.1 | 56.7 | 48.5 | 78.4 | 30.6 | 59.0 |
| Sentence-T5$_{\text{xxl}}$ (Ni et al., 2021b) | 4.8B | 73.4 | 43.7 | 85.1 | 56.4 | 42.2 | 82.6 | 30.1 | 59.5 |
| `text-embedding-ada-002` | NA | 70.9 | 45.9 | 84.9 | 56.3 | 49.3 | 81.0 | 30.8 | 61.0 |
| `text-embedding-3-small` | NA | 73.2 | 46.7 | 85.0 | 56.7 | 51.1 | 81.6 | 31.1 | 62.3 |
| `text-embedding-3-large` | NA | 75.5 | 49.0 | 85.7 | 59.2 | 55.4 | 81.7 | 29.9 | 64.6 |
| E5$_{\text{mistral}}$ (Wang et al., 2023b) | 7B | 78.5 | 50.3 | 88.3 | 60.2 | 56.9 | 84.6 | 31.4 | 66.6 |

## 5.2 Text Embedding Benchmark Results

To evaluate nomic-embed-text-v1 effectiveness as a text encoder, we evaluate it on MTEB (Muennighoff et al., 2023), Jina's Long Context Benchmark (Günther et al., 2024), and LoCo (Saad-Falcon et al., 2024).

### 5.2.1 MTEB Results

MTEB is a general text embedding benchmark released by Muennighoff et al. (2023). It measures text embedding performance across tasks Classification, Clustering, Pair Classification, Reranking, Retrieval, STS, and Summarization.

During evaluation, we add the `classification` prefix to both the query and document for the Classification, Pair Classification, STS, and Summarization tasks. We add the `clustering` prefix to both the query and document for the Clustering task. And we add the `search_query` prefix to the query and `search_document` prefix to the document for the Retrieval task. We truncate all texts to 512 tokens. Additionally, we found better performance by not L2 normalizing the embeddings for the Classification task, similar to the evaluation code released by Wang et al. (2022). For all other tasks, we L2 normalize the embeddings.

The performance of nomic-embed-text-v1 and nomic-embed-text-v1-ablated are broken down by task in Table 4. Compared to similarly sized open-source text embedding models, nomic-embed-text-v1 outperforms all models except BGE-base (Xiao et al., 2023). Additionally, nomic-embed-text-v1 outperforms larger open-source text embedding models like E5 Large v2 (Wang et al., 2022), GTR XXL (Ni et al., 2021a), and Sentence T5 XXL (Ni et al., 2021b).

Table 5: Jina Long Context benchmark results. nomic-embed-text-v1 outperforms jina-embeddings-v2-base and performs similarly to text-embeddings-ada-002.

| Model | Seq | NarrativeQA | WikiCities | SciFact | BigPatent | Avg |
|---|---|---|---|---|---|---|
| jina-embeddings-base-v2 | 128 | 19.6 | 79.9 | 62.1 | 14.4 | 44.0 |
| nomic-embed-text-v1-ablated | 128 | 20.8 | 86.8 | 65.2 | 17.5 | 47.6 |
| nomic-embed-text-v1 | 128 | 20.1 | 90.0 | 65.4 | 18.5 | 48.5 |
| text-embedding-ada-002 | 128 | 25.4 | 84.9 | 68.8 | 16.6 | 48.9 |
| text-embedding-3-small | 128 | 29.5 | 87.5 | 68.8 | 15.0 | 50.2 |
| text-embedding-3-large | 128 | 45.6 | 87.9 | 74.8 | 16.5 | 56.2 |
| jina-embeddings-base-v2 | 512 | 21.3 | 79.3 | 66.7 | 21.9 | 47.3 |
| nomic-embed-text-v1-ablated | 512 | 25.7 | 81.9 | 71.5 | 23.7 | 50.7 |
| nomic-embed-text-v1 | 512 | 23.9 | 88.7 | 70.5 | 25.3 | 52.1 |
| text-embedding-ada-002 | 512 | 25.5 | 84.8 | 72.6 | 23.0 | 51.5 |
| text-embedding-3-small | 512 | 32.2 | 89.0 | 73.2 | 23.6 | 54.5 |
| text-embedding-3-large | 512 | 48.1 | 89.9 | 77.6 | 23.6 | 59.6 |
| jina-embeddings-base-v2 | 8191 | 39.4 | 75.7 | 69.4 | 23.1 | 51.9 |
| nomic-embed-text-v1-ablated | 8191 | 44.0 | 77.4 | 69.1 | 23.6 | 53.5 |
| nomic-embed-text-v1 | 8191 | 37.8 | 84.3 | 70.2 | 24.5 | 54.2 |
| text-embedding-ada-002 | 8191 | 41.1 | 84.7 | 72.7 | 22.5 | 55.3 |
| text-embedding-3-small | 8191 | 47.1 | 89.9 | 73.3 | 22.5 | 58.3 |
| text-embedding-3-large | 8191 | 51.6 | 86.2 | 77.7 | 19.3 | 58.7 |

Table 6: LoCo benchmark results (Saad-Falcon et al., 2024). nomic-embed-text-v1 is the best-performing 100M parameter class unsupervised model. nomic-embed-text-v1 is competitive with the top-performing models in both the 7B parameter class and with models trained in a supervised setting specifically for the LoCo benchmark.

| Model | Seq | Param. | Tau Scr. | Tau Gov. | Tau QMS. | QASP. Tit. Art. | QASP. Abs. Art. | Avg |
|---|---|---|---|---|---|---|---|---|
| M2-Bert (Saad-Falcon et al., 2024) | 2048 | 80M | 81.8 | 94.7 | 58.5 | 87.3 | 95.5 | 83.6 |
| Jina$_{base-v2}$ (Günther et al., 2024) | 2048 | 137M | 87.2 | 97.7 | 35.1 | 95.3 | 99.7 | 83.0 |
| nomic-embed-text-v1-ablated | 2048 | 137M | 83.1 | 97.3 | 49.4 | 97.4 | 99.9 | **85.4** |
| nomic-embed-text-v1 | 2048 | 137M | 86.1 | 96.9 | 47.8 | 96.1 | 99.7 | 85.3 |
| nomic-embed-text-v1 | 4096 | 137M | 89.0 | 97.4 | 45.7 | 95.8 | 99.9 | 85.6 |
| nomic-embed-text-v1-ablated | 4096 | 137M | 89.1 | 97.6 | 49.6 | 97.5 | 99.9 | 86.7 |
| E5$_{mistral}$ (Wang et al., 2023b) | 4096 | 7B | 95.9 | 98.3 | 46.8 | 98.4 | 99.8 | **87.8** |
| M2-Bert (Saad-Falcon et al., 2024) | 8192 | 80M | 94.7 | 96.5 | 64.1 | 86.8 | 97.5 | **87.9** |
| Jina$_{base-v2}$ (Günther et al., 2023) | 8192 | 137M | 93.3 | 98.6 | 40.8 | 95.1 | 99.3 | 85.5 |
| nomic-embed-text-v1-ablated | 8192 | 137M | 92.5 | 97.8 | 47.6 | 96.5 | 99.9 | 86.9 |
| nomic-embed-text-v1 | 8192 | 137M | 90.9 | 97.8 | 44.2 | 94.9 | 99.9 | 85.5 |
| text-embedding-ada-002 | 8192 | N/A | 37.3 | 44.3 | 7.30 | 85.1 | 89.7 | 52.7 |
| text-embedding-3-small | 8192 | N/A | 92.2 | 97.7 | 27.4 | 95.9 | 98.9 | 82.4 |
| text-embedding-3-large | 8192 | N/A | 88.0 | 93.6 | 25.5 | 93.2 | 96.8 | 79.4 |

Compared to closed-source models, nomic-embed-text-v1 outperforms text-embedding-ada-002 and text-embedding-3-small on average and notably the Retrieval task. nomic-embed-text-v1 is the only open-source long-context text embedding model to outperform text-embedding-ada-002 and text-embedding-3-small on MTEB.

nomic-embed-text-v1-ablated unsurprisingly performs worse than nomic-embed-text-v1 and other open-source text embedding models that finetune on the training sets of BEIR like BGE-base and GTE-base. However, nomic-embed-text-v1-ablated still outperforms text-embedding-ada-002 and jina-embeddings-base-v2 and is competitive with E5-base v2. jina-embeddings-base-v2 is trained on similar datasets to nomic-embed-text-v1-ablated yet nomic-embed-text-v1-ablated outperfoms jina-embeddings-base-v2 on MTEB.

### 5.2.2 Long Context Results

However, as noted in Günther et al. (2024), MTEB has very few datasets that include long sequences. To evaluate nomic-embed-text-v1's performance on longer sequences, we consider two additional benchmarks: the Jina Long Context Dataset (Günther et al., 2024) as well as the LoCo benchmark from Saad-Falcon et al. (2024). Although nomic-embed-text-v1 was trained with a max sequence length of 2048, we are able to use length extrapolation techniques proposed in emozilla (2023); Peng et al. (2023).

For texts longer than 2048, the max sequence length nomic-embed-text-v1 was trained on, we employ Dynamic NTK Interpolation as described in Equation 5. We set $\alpha$ to 2.

### 5.2.3 JinaAI Long Context Benchmark

The Jina Long Context Benchmark (Günther et al., 2024) evaluates on 4 datasets across Retrieval and Clustering; namely, NarrativeQA (Günther et al., 2024), WikiCites [7], SciFact (Wadden et al., 2020), and BigPatent [8] (Sharma et al., 2019). Similar to Günther et al. (2024), we report the V-scores and NDCG@10 for the clustering and retrieval datasets respectively. We evaluate all models at sequence length 128, 512, and 8191. For nomic-embed-text-v1 and nomic-embed-text-v1-ablated on NarrativeQARetrieval and Scifact, we use the `search_query` and `search_document` prefixes for the query and document respectively. For BigPatentClustering and WikiCities, we use the `clustering` prefix for both the query and document. Results are presented in Table 5.

Numbers for `text-embedding-ada-002` and `jina-embeddings-base-v2` are taken from (Günther et al., 2024).

Across all context lengths, nomic-embed-text-v1 outperforms jina-embeddings-v2-base. When evaluated on shorter sequence lengths, nomic-embed-text-v1 performs similarly to text-embedding-ada-002 but is outperformed at 8k context. Additionally, nomic-embed-text-v1-ablated outperforms jina-embeddings-v2-base, but underperforms nomic-embed-text-v1.

However, nomic-embed-text-v1 underperforms text-embedding-ada-002, text-embedding-3-small, and text-embedding-3-large. Without any information on training data or architecture for the closed-source models, it's unclear why the gap exists. It is also surprising to see text-embedding-3-large performance decrease as sequence length increases from 512 to 8191 while performance increases for text-embedding-3-small and text-embedding-002-ada.

Similar to results in Günther et al. (2023), we see lower performance in WikiCities across models as sequence length increases suggesting the task may not be a good measure of long context embedding performance.

### 5.2.4 LoCo Benchmark

The LoCo Benchmark consists of 5 retrieval datasets: 3 datasets from Shaham et al. (2022) and 2 from Dasigi et al. (2021). Similar to the other retrieval evaluations, we use the `search_query` and `search_document` prefixes for the query and document respectively. We evaluate nomic-embed-text-v1 and jina-embeddings-base-v2 at sequence length 2048, 4096, and 8192. We additionally include results from Saad-Falcon et al. (2024) as well even though the model was finetuned on training sets of these datasets. Results are presented in Table 6. We include the QASPER Abstract Articles dataset for completeness, but would like to highlight that many models seem to oversaturate the benchmark and may not be representative of long-context performance.

---

[7]https://huggingface.co/datasets/jinaai/cities_wiki_clustering
[8]https://huggingface.co/datasets/jinaai/big-patent-clustering

At 2048 sequence length, nomic-embed-text-v1 and nomic-embed-text-v1-ablated outperform jina-embeddings-v2-base. At a 4096 sequence length nomic-embed-text-v1 and nomic-embed-text-v1-ablated is able to perform similarly to E5 Mistral, a model $\approx$ 70x bigger on all tasks except Tau Scrolls.

Both nomic-embed-text-v1 variants outperform text-embedding-ada-002 and text-embedding-3-small and perform similarly to jina-embeddings-v2-base at 8192 sequence length. Interestingly, nomic-embed-text-v1-ablated outperforms nomic-embed-text-v1 and jina-embedding-base-v2 suggesting that the BEIR training data may be orthogonal to the LoCo tasks.

## 6 Conclusion

We release the first fully open-source long context text embedding model that surpasses OpenAI's text-embedding-Ada-002 and text-embedding-003-small performance on both sort and long context benchmarks. We release the model weights and training code under a permissible license as well as the recipe, including data, to reproduce the model. As of this writing, nomic-embed has garnered over 14 million downloads on the Hugging Face model hub, underscoring the widespread demand for performant open source model recipes.

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

## A  Training Resources

Full training of nomic-embed-text-v1 can be conducted in a single week on one 8xH100 node. Masked language modeling of nomic-bert-2048 takes roughly 4 days. Contrastive pretraining lasts 3 and a half days. Contrastive finetuning takes one hour. We encourage the reader to initialize from our nomic-bert-2048 or Unsupervised Constrastive checkpoints, released under the same license as nomic-embed-text-v1.

Table 7: Weakly Unsupervised Dataset Distribution

| Dataset | Datapoints | % Dataset |
|---|---|---|
| Reddit [a] | 64,978,944 | 0.28 |
| PAQ (Lewis et al., 2021b) | 52,953,088 | 0.23 |
| Amazon Reviews (Ni et al., 2019) | 38,682,624 | 0.16 |
| S2ORC Title Abstract (Lo et al., 2020) | 35438592 | 0.15 |
| WikiAnswers (Fader et al., 2014) | 9,912,320 | 0.04 |
| S2ORC Citation Titles (Lo et al., 2020) | 7,585,792 | 0.03 |
| S2ORC Abstract Citation (Lo et al., 2020) | 7,503,872 | 0.03 |
| S2ORC Abstract Body (Lo et al., 2020) | 6,389,760 | 0.03 |
| Wikipedia Title Body (Foundation) | 6,078,464 | 0.03 |
| Gooaq (Khashabi et al., 2021) | 1,245,184 | 0.01 |
| Codesearch (Husain et al., 2019) | 835,584 | <.01 |
| AGNews (Zhang et al., 2016) | 409,600 | <.01 |
| CCNews (Hamborg et al., 2017) | 344,064 | <.01 |
| NPR [b] | 344,064 | <.01 |
| CNN (See et al., 2017) | 278,528 | <.01 |
| Yahoo Title-Answer [c] | 262,144 | <.01 |
| AmazonQA (Gupta et al., 2019) | 212,992 | <.01 |
| Yahoo Title-Question [d] | 196,608 | <.01 |
| Sentence Compression (Filippova & Altun, 2013) | 163,840 | <.01 |
| YahooQA [e] | 131,072 | <.01 |
| ELI5 (Fan et al., 2019) | 98,304 | <.01 |
| Altlex (Hidey & McKeown, 2016) | 98,304 | <.01 |
| Wikihow (Koupaee & Wang, 2018) | 81,920 | <.01 |
| SimpleWiki (Coster & Kauchak, 2011) | 81,920 | <.01 |
| StackExchange Duplicate Questions [f] | 65,536 | <.01 |
| StackExchange Title Body [g] | 65,536 | <.01 |
| StackExchange Body Body [h] | 65,536 | <.01 |
| Quora Duplicate Questions [i] | 32,768 | <.01 |
| SQuAD (Rajpurkar et al., 2016) | 16,384 | <.01 |
| Total | 234,553,344 | 1 |

[a] https://huggingface.co/datasets/sentence-transformers/reddit-title-body
[b] https://files.pushshift.io/news/
[c] https://www.kaggle.com/soumikrakshit/yahoo-answers-dataset
[d] https://www.kaggle.com/soumikrakshit/yahoo-answers-dataset
[e] https://www.kaggle.com/soumikrakshit/yahoo-answers-dataset
[f] https://data.stackexchange.com/apple/query/fork/1456963
[g] https://data.stackexchange.com/apple/query/fork/1456963
[h] https://data.stackexchange.com/apple/query/fork/1456963
[i] https://quoradata.quora.com/First-Quora-Dataset-Release-Question-Pairs

## B    Pretraining Dataset Distribution

Weakly-supervised contrastive pretraining datasets are detailed in Table 7. This is the number of datapoints per source after consistency filtering.

