# OpenReview forum: "Nomic Embed: Training a Reproducible Long Context Text Embedder"
_TMLR — Accepted by TMLR_

### Review · Reviewer_2U5B · 2024-12-10

**Summary Of Contributions:**

The key contributions of the paper are the introduction of anon-embed-text-v1, a fully reproducible, open-source long-context text embedding model with an 8192-token context length. The model outperforms OpenAI’s text-embedding-ada-002 and text-embedding-3-small on both short-context (MTEB) and long-context (LoCo, Jina Long Context Benchmark) benchmarks, despite having only 137M parameters. The paper provides complete transparency by releasing model weights, training code, and curated training data. The training pipeline includes masked language modeling, weakly-supervised contrastive pretraining, and supervised contrastive finetuning, utilizing techniques like consistency filtering and hard negative mining. Additionally, the model leverages architectural optimizations such as RoPE, SwiGLU, and FlashAttention for efficiency and scalability.

**Audience:**

No

**Claims And Evidence:**

Yes

**Requested Changes:**

A few more details about the  negative mining and consistency filtering process, and the metric of choosing such filtering and mining hyperparameters.

A few more details determining corpus size of the second and third stage and when you would think the stage2 is good enough for starting stage 3 training.

**Strengths And Weaknesses:**

Strength:

The paper does comprehensive experiments on training text embeddings, which covers from initial model pretraining, weakly supervised data and supervised data. It also performs broad evaluation on the most used benchmarks regarding different aspects of text embeddings.  The paper outputs a better BERT model with modifications from recent LLM updates in structure and surpass the performance of original BERT.

The paper also divides the contrastive learning into weakly and full supervised training with details about how the data was choosen and filtered, for example, consistency filtering and negative sample mining, achieving better performance with a relative smaller computation cost in inference.

The paper also promises code and data for this training pipeline, which iof cource benenfits the community.


Weakness:

The paper leverages masked language model (training from scratch) as a base model for the text representation learning. I am concerned that the method may not be able to take advantage of the fast-pacing improvement from the generative text modeling. For example, this base model is limited to 2048 tokens in length, total training tokens, model size, etc. In the meantime, there has already been tiny but strong open-sourced gpt models which is trained on longer sentences, consumes more tokens and reasonable in model size. Unless there's a strong reason for BERT, I would suggest transferring to a tiny gpt as the base structure.


Questions:

For the data filtering, how did you decide the hyperparameters of negative mining and consistency filtering, is there any metric to determine this hyperparameters?

---

> ### Author Response · Authors · 2024-12-28
> **Response to Reviewer 2U5B**
>
> We would like to thank the reviewer for their insightful questions and feedback.
>
> Below, you will find our responses to the questions and feedback
>
> >The paper leverages masked language model (training from scratch) as a base model for the text representation learning. I am concerned that the method may not be able to take advantage of the fast-pacing improvement from the generative text modeling. For example, this base model is limited to 2048 tokens in length, total training tokens, model size, etc. In the meantime, there has already been tiny but strong open-sourced gpt models which is trained on longer sentences, consumes more tokens and reasonable in model size. Unless there's a strong reason for BERT, I would suggest transferring to a tiny gpt as the base structure.
>
> **Response**: Thank you for this insightful suggestion about leveraging GPT-style architectures. We appreciate the points raised about potential benefits of tiny GPT models, particularly regarding sequence length and training scale.
>
> We specifically targeted a ~100M parameter model to enable deployment in low-resource environments.
>
> Additionally, we experimented with contrastively pretraining and finetuning Pythia 1B but found no or small performance improvements over Bert-base. The 10x increase in surprisingly didn’t lead to a large increase in performance.
>
> That said, we agree this is an interesting direction for future work, particularly given recent advances in small decoder models. Specifically, exploring how newer decoder architectures could be efficiently adapted for embedding models is an interesting and under explored direction.
>
> >For the data filtering, how did you decide the hyperparameters of negative mining and consistency filtering, is there any metric to determine this hyperparameters? A few more details about the negative mining and consistency filtering process, and the metric of choosing such filtering and mining hyperparameters.
>
> **Response**: Thank you for these thoughtful questions about our consistency filtering approach. We appreciate the opportunity to clarify our methodology:
> Regarding model selection and hyperparameter choices: Our approach was primarily empirical rather than metric-driven. During initial experiments, we observed that all-MiniLM-L6-v2 was systematically discarding valuable retrieval pairs, particularly those with low lexical overlap but high semantic similarity. Initial experiments produced models that performed worse on retrieval tasks for a similar number of steps. This led us to explore GTE-base, which qualitatively produced better filtering results due to its broader training distribution across diverse datasets.
>
>
> For the choice of k=2 in top-k filtering: This was determined through manual inspection of filtered results rather than an explicit optimization metric. We experimented with both threshold-based filtering and different k values, but found through qualitative analysis that:
> * k=1 was too restrictive, removing many valid pairs
> * k>2 retained too many pairs and led to a dataset that was outside of our compute budget
> * Threshold-based approaches lead to models that struggled on retrieval tasks
>
>
> We acknowledge this is a limitation of our work - developing quantitative metrics for optimizing these filtering parameters would be valuable future work.
>
>
> For hard negative mining, we have clarified the section in the paper to better describe how negative mining was done. In short, the top 20 negatives, excluding the positive document, were selected from the corpus using GTE-Base. During training, we sample from the 20 mined negatives instead of taking the first N negatives due to potential false negatives. We additionally added a table outlining the data distribution of the finetuning dataset
>
> > A few more details determining corpus size of the second and third stage and when you would think the stage2 is good enough for starting stage 3 training.
>
> **Response**: The corpus size for stage 2 training is mentioned in section 4.2.1 and the full dataset distribution can be found in Appendix B (Table 7). The original noisy dataset is ~900 million samples and after consistency filtering is 235 million samples. We added a table outlining the 1.6 million sample stage 3 dataset size and distribution as well in section 4.2.4.
>
> The question of when stage 2 pretraining has reached an optimal point to begin stage 3 finetuning is an interesting future direction. We ran initial experiments training for multiple pretraining epochs but found diminishing returns and it’s unclear to us if data quantity or quality is more important for a high performant text embedding model. Future work similar to Cramming [1] and Scaling Data-Constrained Language Models [2] applied to text embeddings would be to the community.
>
> [1]: Cramming: Training a Language Model on a Single GPU in One Day  (Geiping & Goldstein, 2022)
>
>
> [2]: Scaling Data-Constrained Language Models (Muennighoff et al 2023)

---

### Review · Reviewer_nt2w · 2024-12-14

**Summary Of Contributions:**

This technical report presents Anon Embed, the first fully open and reproducible long-context embedder. It provides a comprehensive overview of the model architecture, training algorithm, and selected hyperparameters. The embedder is trained using a combination of masked language modeling, weakly supervised contrastive pretraining, and contrastive fine-tuning. To support scaling to 8192 tokens during inference, rotary positional embeddings and interpolation techniques are employed. Extensive experiments validate the model's performance, demonstrating its effectiveness.

**Audience:**

Yes

**Broader Impact Concerns:**

The paper does not present any obvious concerns.

**Claims And Evidence:**

Yes

**Requested Changes:**

1. The contribution of the work could be enhanced by training and open-sourcing a larger model with improved performance.
2. The presentation of result tables could be more organized and clear. For instance, the order of models and the usage of bolding is somewhat confusing, making it challenging to extract information efficiently.

**Strengths And Weaknesses:**

Strengths:
- The paper fully open-sources its implementation details, promoting transparency and fostering advancements in related research and applications.
- Extensive experiments demonstrate that the proposed embedder achieves performance comparable to state-of-the-art models.
- In long-context scenarios, the anon-embed-text-v1-ablated model outperforms various baseline models, showcasing its effectiveness in handling extended sequences.

Weaknesses:
- The model's performance on relatively short sequences falls short compared to baseline models.
- With a relatively small number of parameters, the model shows clear disadvantages when compared to larger models such as E5.

---

> ### Author Response · Authors · 2024-12-28
> **Response to Reviewer nt2w**
>
> We would like to thank the reviewer for their valuable feedback and comments.
>
> > The model's performance on relatively short sequences falls short compared to baseline models. With a relatively small number of parameters, the model shows clear disadvantages when compared to larger models such as E5.
>
> **Response**: In Table 3, anon-embed-text outperforms all bert-base sized models except BGE-Base on short sequence tasks while having the ability to extend to sequence lengths longer than 512 tokens. Additionally, Table 3 shows that anon-embed-text-v1 outperforms larger models like E5 Large v2 on average and GTE Large on retrieval while being 2x smaller.
>
> >The contribution of the work could be enhanced by training and open-sourcing a larger model with improved performance.
>
> **Response**: While scaling to larger models is a possible direction for future work, our current results demonstrate that architectural and data improvements can achieve competitive performance without requiring larger models. As shown in Table 3, our base-sized model already outperforms several larger models like E5 Large v2 and GTE Large, suggesting that simply scaling up parameters may not be the most efficient path forward. Our focus on achieving strong performance with a smaller, more efficient architecture provides practical value for resource-constrained applications.
>
> >The presentation of result tables could be more organized and clear. For instance, the order of models and the usage of bolding is somewhat confusing, making it challenging to extract information efficiently.
>
> **Response**: Thank you for highlighting the lack of clarity! We have reordered the columns and put Seq first and reordered the rows in increasing sequence length order and added more details to the description. For table 3, we added a line to distinguish between bert-base and bert-large sized models, ordered models in increasing MTEB score, and updated the description. We reordered the rows in ascending average score across sequence lengths and added more details to the description. We reordered the rows in Table 4 to be in increasing order of model parameters and bolded the highest score at each sequence length.

---

### Review · Reviewer_FaWz · 2024-12-26

**Summary Of Contributions:**

The paper provides a comprehensive account of developing the nomic-embed-text-v1 model, an open-source English text embedding model with an 8192-token context length. The training process comprises three primary stages:
1. Masked Language Modeling (MLM) Pretraining: The model, nomic-bert-2048, was pretrained using a masked language modeling objective on datasets such as BooksCorpus and Wikipedia.
2. Unsupervised Contrastive Pretraining: Following MLM pretraining, the model underwent unsupervised contrastive learning using a large corpus of weakly paired data, including question-answer pairs from forums and title-body pairs from reviews.
3. Supervised Fine-tuning: The final stage involved fine-tuning the model on smaller, high-quality labeled datasets, such as MSMarco, to enhance its performance on specific tasks.

Additionally, the authors provide detailed methodologies for data collection and filtering, ensuring the model’s reproducibility and performance. They have released the training code, model weights, and a curated dataset of 235 million text pairs under an Apache 2 license, facilitating full replication of the model.

**Audience:**

Yes

**Broader Impact Concerns:**

The authors does not delve into potential ethical, societal implications associated with the deployment of the data and models.

**Claims And Evidence:**

Yes

**Requested Changes:**

Except the weaknesses mentioned above,
The Consistency Filtering section is important but a little bit hard to follow, e.g. Offering brief descriptions of the all-MiniLM-L6-v2 and gte-base models to contextualize their selection.

**Strengths And Weaknesses:**

The paper presents nomic-embed-text-v1, an open-source English text embedding model with an 8192-token context length. Unlike typical sentence embedding models that build upon pretrained decoder-based language models, this work initiates from a masked language model, emphasizing data quality and quantity. The model demonstrates comparable performance on both short and long documents, contributing to advancements in representation models. Its relatively small parameter size is advantageous for applications like retrieval, where efficiency is crucial.

While the paper is commendable, certain sections could benefit from refinement. For instance, Section 3.4 covers too much general knowledge that doesn’t directly reflect the authors’ contributions; condensing this part could make room for more discussion on novel findings, such as the impact of varying the parameter θ. Similarly, if Section 3.5.2’s discussion on NTK-Aware Scaling isn’t utilized in the experiments, it might be more appropriate for the supplementary materials. Additionally, enhancing table designs and descriptions would improve readability. In Table 2, reordering columns to place ‘Model’ and ‘Seq’ first could help readers quickly identify key differentiators. In Section 4.2.1, the raw version data uses words "472 million", the filter data uses words "~235M pairs".

Overall, the paper makes a significant contribution, and addressing these suggestions could enhance its clarity and impact.

---

> ### Author Response · Authors · 2024-12-28
> **Response to Reviewer FaWz**
>
> We thank the reviewer for their positive feedback and address their concerns below.
>
> > While the paper is commendable, certain sections could benefit from refinement. For instance, Section 3.4 covers too much general knowledge that doesn’t directly reflect the authors’ contributions; condensing this part could make room for more discussion on novel findings, such as the impact of varying the parameter θ. Similarly, if Section 3.5.2’s discussion on NTK-Aware Scaling isn’t utilized in the experiments, it might be more appropriate for the supplementary materials. Additionally, enhancing table designs and descriptions would improve readability. In Table 2, reordering columns to place ‘Model’ and ‘Seq’ first could help readers quickly identify key differentiators. In Section 4.2.1, the raw version data uses words "472 million", the filter data uses words "~235M pairs".
>
> **Response**: Thank you for highlighting the sections that could use refinement.  We have updated all places and replaced M with million and B with billion. We have reordered the columns and put Seq first and reordered the rows in increasing sequence length order and added more details to the description. For table 3, we added a line to distinguish between bert-base and bert-large sized models, ordered models in increasing MTEB score, and updated the description. We reordered the rows in ascending average score across sequence lengths and added more details to the description. We reordered the rows in Table 4 to be in increasing order of model parameters and bolded the highest score at each sequence length. We additionally added more descriptive table titles. We have shortened section 3.4 and only kept the section on RoPE length extrapolation as it is later used to describe how we employ length extrapolation for long context tasks.
>
> > The Consistency Filtering section is important but a little bit hard to follow, e.g. Offering brief descriptions of the all-MiniLM-L6-v2 and gte-base models to contextualize their selection
>
> **Response**: Thank you for this feedback about the clarity of our Consistency Filtering section. We have revised this section to better contextualize our model selection decisions. Specifically, we:
>
> 1. Added parameter counts for both models (22M for all-MiniLM-L6-v2 vs 109M for gte-base) to provide concrete size comparison
> 2. Explained our rationale for choosing gte-base over all-MiniLM-L6-v2: we found the smaller model regularly discarded valid retrieval pairs that had low lexical overlap but high semantic similarity
>
> We believe these changes make the filtering methodology and model selection process more transparent and easier to follow.

---

### Decision · Action_Editor_nwtt · 2025-02-03

**Recommendation:** Accept with minor revision

**Comment:**

The reviewers suggestion several revisions, please consider it include them. The writing style of the tables should be also improved.

**Audience:**

Text representations and embeddings are fundamental topics in NLP and widely used in industry practices. The main appeal of this work to the audience lies in its open-source nature, which includes reproducible results, open weights, and open data.

**Claims And Evidence:**

The paper presents a fully open-source English text embedding model, emphasizing data quality and quantity. The model demonstrates comparable performance on both short and long documents. Its relatively small parameter size is advantageous in terms of efficiency.